# Assessment of vaccine perception and vaccination intention of Mpox infection among the adult males in Bangladesh: A cross-sectional study findings

Md. Rabiul Islam[1]*, Md. Anamul Haque[1], Bulbul Ahamed[1], Md. Tanbir[1], Md. Robin Khan[1], Saba Eqbal[1], Md. Ashrafur Rahman[2], Mohammad Shahriar[1], Mohiuddin Ahmed Bhuiyan[1]

1 Department of Pharmacy, University of Asia Pacific, Dhaka, Bangladesh, 2 Department of Pharmaceutical Sciences, Wilkes University, Wilkes Barre, PA, United States of America

☯ These authors contributed equally to this work.

* robi.ayaan@gmail.com

**Data Availability Statement:** All relevant data are within the paper and its Supporting Information files.

## Abstract

### Background

Mpox (monkeypox) infection has become a global concern for healthcare authorities after spreading in multiple non-endemic countries. Following the sudden multi-country outbreak of Mpox, the World Health Organization (WHO) declared a public health emergency of international concern. We do not have any vaccines approved for the prevention of Mpox infection. Therefore, international healthcare authorities endorsed smallpox vaccines for the prevention of Mpox disease. Here we intended to perform this cross-sectional study among the adult males in Bangladesh to assess the Mpox vaccine perception and vaccination intention.

### Methods

We conducted this web-based survey among the adult males in Bangladesh from September 1, 2022, to November 30, 2022, using Google Forms. We assessed the Mpox vaccine perception and vaccination intention. We performed a chi-square test to compare vaccine perception and vaccination intention levels. Also, we performed multiple logistic regression analyses to determine the association between the study parameters and the sociodemographic profile of the participants.

### Results

According to the present study, the Mpox vaccine perception was high among 60.54% of the respondents. Also, 60.05% of respondents showed medium vaccination intention. Mpox vaccine perception and vaccination intention were strongly associated with the sociodemographic profiles of the participants. Furthermore, we discovered a significant association

**Funding:** The author(s) received no specific funding for this work.

**Competing interests:** The authors have declared that no competing interests exist.

between the level of education and vaccination intention among the respondents. Also, age and marital status played a role in the Mpox vaccine perception and vaccination intention.

## Conclusion

Our findings showed a significant association between sociodemographic characteristics and the Mpox vaccine perception/vaccination intention. Along with the country's long experience in mass immunization, campaigns about Covid-19 vaccines and high vaccination rates might play a role in Mpox vaccine perception and vaccination intention. We recommend more social awareness and educational communications or seminars for the target population to bring more positive changes in their attitude towards Mpox prevention.

## Introduction

The orthopoxvirus genus of the poxviridae family contains the enclosed double-stranded DNA virus known as the Mpox (monkeypox) virus. The Mpox virus is divided into two genetic clades: the Central African (Congo Basin) clade and the West African clade [1, 2]. On November 28, 2022, the World Health Organization (WHO) renamed the disease as Mpox to avoid stigma and racism [3]. Mpox is a zoonotic disease that can be spread to people by direct contact with primates and several rodent species [1]. In 1959, a shipment of sick monkeys from Singapore to a Danish research center led to the first isolation and identification of the Mpox virus. The virus was identified in a child in the Democratic Republic of the Congo who was thought to have smallpox, nevertheless, and it was the first known occurrence of the disease in a human being [4, 5]. However, when it comes to rates of complications, case fatalities, and scarring, Mpox is less severe than smallpox [5–7]. There have been numerous epidemics of Mpox that were spread from person to person [8, 9]. Transmission through respiratory droplets and direct contact with lesions of an infected person is causing the present outbreak. According to recent data, there is also the possibility of sexual transmission of Mpox [4, 10, 11]. In West African places where there is frequent contact between people and wild animal reservoirs and where there is evidence that the infectious attack rate is rising, Mpox has been identified as a growing public health hazard [5]. The Mpox virus replicates at the site of inoculation after viral entrance by any route (intradermal, nasopharynx, and oropharynx) and subsequently spreads to nearby lymph nodes. According to studies, the incubation period lasts between 5 and 21 days, and signs and symptoms last between 2 and 5 weeks. Fever, headaches, myalgia, lethargy, and lymph node swellings are some of the signs and symptoms that precede the sickness. Rashes of various sizes start from 1 to 5 days after the start of the fever, first on the face, and may or may not spread to other areas of the body [4, 5]. In 2022, there is now a widespread Mpox outbreak across several nations on various continents, primarily affecting the population of men who have sex with men (MSM), bisexual, and gay. In Spain, 99% of the 595 confirmed cases of Mpox were found to be in the MSM population, with the lesions mostly affecting the genital, perianal, and perineal regions. However, this may happen as men often hunt and interact with wild animals [4, 12]. The increase in cases of human Mpox highlights the value of prevention strategies, as well as early screening and identification [13].

Mpox has recently become a global health problem, with the WHO reporting over 80,000 proven and suspected cases in over 90 countries across Europe, the Western Pacific, Southeast Asia, the Americas, and the Eastern Mediterranean. Furthermore, the number of instances likely to grow indefinitely [3, 14, 15]. It has become vital to take preventive measures against

Mpox infection by creating social awareness and managing supportive care and symptomatic treatment [16, 17]. Massive health education campaigns are required to raise public awareness and advise infected individuals to remain in isolation, wear protective clothing and surgical masks, as well as refrain from close contact with healthy individuals [9, 18]. Also, safety precautions and quarantine must be maintained for any healthy animal that may have come into close contact with an infected animal and monitored for signs of Mpox for 30 days. These social norms can help prevent the transmission of the Mpox virus from human-to-human [1, 14, 19]. While caring for Mpox patients, healthcare workers must maintain strict hand washing protocol, handle contaminated medical equipment cautiously, sterilize environmental surfaces, and dispose of laundry and garbage appropriately [19]. Individuals with infection should cover the lesions as long as possible until all lesion has spontaneously fallen off to form a new skin layer. Presently, there is no specific effective therapy other than early vaccination following viral exposure [20]. The Centers for Disease Control and Prevention (CDC) states that disease development may halt by getting vaccinated within four days of exposure, and disease severity may fall by injecting a vaccine within 14 days [4]. However, Mpox vaccines are not yet present in Bangladesh and several other countries. Scientists have recommended some smallpox vaccines for preventing the Mpox outbreak as Mpox's clinical presentation is similar to that of smallpox. According to some research studies, several animal models were vaccinated with smallpox vaccines and provided protection against a variety of orthopoxvirus [21]. Besides, it has been estimated that smallpox vaccines (ACAM2000 and JYNNEOS) provide 85% cross-protection against Mpox disease. However, the risks linked with JYNNEOS are lower than those of ACAM2000. Both smallpox vaccines: ACAM2000, a live-attenuated replicating vaccine, and JYNNEOS, a replication-deficient MVA vaccine, are FDA-approved vaccines to fight against the Mpox virus [14, 22, 23]. Thus, vaccination is the key preventive measure for the Mpox outbreak.

Vaccine hesitancy is recognized as a hazard to public health by the WHO [24]. Misconceptions regarding vaccination abound, and, more crucially, vaccine reluctance is on the rise. In particular, the impact size of vaccinating others is substantially bigger than vaccinating oneself [25, 26]. The fear of vaccinations raises vaccine refusal rates, which increases the chances of illness and death from diseases that may be prevented by vaccines [27]. Prior research has suggested the importance of improving people's perceptions of their ability to manage their vaccination behavior [28]. Both internal and external factors frequently influence these control-based attitudes, such as the availability of information, resources, willpower, skills, opportunities, sociodemographic differences, vaccine attributes, and speed of vaccine development [26, 29]. However, to roll out vaccines in any country, perception must be good among the potential recipients, and a positive attitude or positive intention must be present toward vaccination [30]. Affective attitude is triggered by emotions and frequently assessed on a continuum of good and bad emotions (e.g., desirable vs undesirable). On the other hand, cognitive attitude is generated by logical judgements and frequently quantified on a continuum of advantages and disadvantages (e.g., safe vs unsafe). Earlier studies revealed that boosting vaccination intention is influenced by enhanced cognitive attitude rather than affective attitude [26]. Furthermore, attention must be drawn to create awareness, willingness, and high positive perceptions among humans for injecting the smallpox vaccine to prevent numerous cases of the Mpox disease. Building public trust in immunization programs requires effective communication between government officials and the public clearly and consistently. This involves describing how vaccines are developed and how they receive regulatory approval based on their safety and effectiveness. Additionally, effective advertisements must emphasize the value of widespread vaccination coverage for achieving community immunity and the duration of protection from vaccinations [31]. The whole purpose of the research study was to determine

the perception and intention of the Mpox vaccine among the Bangladeshi adult male population.

## Materials and methods

### Study design and participants

We conducted a cross-sectional study among the Bangladeshi adult males using Google survey tools from September 1, 2022, to November 30, 2022. A total of 419 students filled out the survey form, but we were only able to analyze 408 responses because 11 of them contained incomplete or inaccurate information. Respondents received the survey form in-person and online via various social media platforms. Only adult males were invited to participate in this study of ages ranging from 18 to 45 years as this age is vulnerable to the Mpox virus. Moreover, several data studies around the world showed 98% of confirmed cases of Mpox disease among 18–45 years of males. Therefore, we excluded females of all ages. All participants who were male Bangladeshi ethnicity and residents provided electronic consent. Participation in this study was entirely voluntary.

### Estimations

The goal was to estimate the relationships between the participants' sociodemographic profile, vaccine perception, and vaccination intention. Accordingly, we prepared a self-reported semi-structured questionnaire in English and translated it into Bangla for better understanding and clarity. We then conducted a pilot study among a small, randomly chosen group of participants to confirm that the survey questions were comprehensible and readable.

### Assessment tools

We used three different tools to assess the participants' sociodemographic profile, vaccine perception, and vaccination intention. The sociodemographic variables included age, marital status, gender, occupation, sexual orientation, body mass index, living status, smoking habit, information about sexual partner and the impact of Covid-19 pandemic on their personal and social life. Simultaneously, we asked six questions regarding vaccine perception and three regarding vaccination intention. Each individual responded to the nine questions under five options: totally disagree (0), disagree (1), neutral (2), agree (3), and totally agree (4).

### Statistical analysis

We processed and analyzed the data with the help of Statistical Packages for Social Sciences, SPSS (version 25.0). Furthermore, we employed descriptive statistical techniques to analyze the demographics and variations among the respondent. Subsequently, for data editing, coding, sorting, classification, and tabulation, we used Microsoft Excel 2016. The chi-square test helped to determine the consequences of the Mpox outbreak with vaccination and other relevant parameters. Regression analysis was performed to measure the association of vaccine perception and vaccination intention with different sociodemographic profiles of the participants. A p-value of 0.05 or below was considered significant in the test results.

### Ethics

The protocol was approved by the Research Ethics Committee, University of Asia Pacific, Dhaka, Bangladesh (Ref: UAP/REC/2022/l09(S1). We conducted this study following the principles stated in the Declaration of Helsinki. Also, we obtained informed electronic consent from all the participants.

## Results

We displayed the demographics of the research participants in Table 1. Almost half of the respondents were between (18–25) years of age and had a normal BMI of 65%. Moreover, among the 408 participants, 61% were unmarried, living with family (76%), non-smokers (68%), and 41% were graduates or above. We discovered that nearly 99% of the participants were heterosexual, and 93% belonged to a lower economic class. Also, most of them (73%) had no impact of Covid-19 on their life and 60% had no sexual partners.

According to Fig 1, the prevalence of Mpox vaccine perception among the respondents was observed to be (i) low (1.47%), (ii) medium (37.99%), and (iii) high (60.54%).

Also, the prevalence of vaccination intention among the participants was observed to be (i) low (3.68%), (ii) medium (60.05%), and (iii) high (36.27%). We have noticed a correlation between various sociodemographic factors with vaccine perception and vaccination intention. The frequency of vaccine perception is higher in (i) individuals who are graduates/above versus higher secondary (51% versus 27.5%, p<0.001), (ii) unmarried versus married (69.6% versus 30.4, p<0.001), (iii) aged (years) 18–25 versus 26–35 (54.7% versus 38.4%, p<0.001), (iv) low versus medium economic status (93.9% versus 3.2%, p<0.001), (v) BMI of (18.5–25) kg/m$^2$ versus above 25 kg/m$^2$ (64.8% versus 24.7%, p<0.001), (vi) heterosexual versus homosexual (99.2% versus 0.4%, p<0.001), (vii) having no sexual partner versus having one (68.9% versus 27.9%, p<0.001), (viii) living with a family versus without family (70.4% versus 29.6%, p<0.001), and (ix) residing in an urban area versus rural area (75.3% versus 24.7%, p<0.001), respectively. Also, 67.2% of the participants disagreed with having an impact of COVID-19 on their lives but the rest 32.8% agreed (Table 2).

We used the binary logistic regression method to measure the correlations between various parameters and vaccine perception (Table 3).

Moreover, the proportion of vaccination intention is at a medium level for (i) BMI (kg/m$^2$) of 18.5–25 versus above 25 (64.9% versus 28.6%, p<0.001), (ii) aged (years) 18–25 versus 26–35 (50.2% versus 36.7%, p<0.001), (iii) individuals who are graduate/above versus higher secondary (35.5% versus 26.5%, p<0.001), (iv) low versus medium economic status (92.2% versus 6.6%, p<0.001), (v) non-smoker versus smoker (64.1% versus 35.9%, p<0.001), (vi) student versus service holder (34.3% versus 29.8%, p<0.001), and (vii) having no sexual partner versus having one (56.8% versus 37.1%, p<0.001), respectively. Additionally, 23.3% of the people involved in the survey agreed that COVID-19 had an impact on their lives, while the remaining 76.7% disagreed (Table 4).

Similar to Mpox vaccine perception, we also assessed the associations between vaccination intention and sociodemographic parameters (Table 5). Respondents with BMI (kg/m$^2$) above 25 had 1.32 times more low vaccination intention than below 18.5 (OR = 1.32, 95% CI = 0.14 to 12.04, p = 0.808). Participants residing in the urban area had high vaccination intention of 2.79 times greater than those in rural area (OR = 2.79, 95% CI = 0.39 to 20.03, p = 0.309). Also, the likelihood of having low vaccination intention with no sexual partner was half than with one partner (OR = 0.50, 95% CI = 0.07 to 3.76, p = 0.502).

Respondents who were (26–35) years old had 1.31 times more high vaccine perception than (36–45) years old (OR = 1.31, 95% CI = 0.50 to 3.38, p = 0.585). The likelihood of having high vaccine perception with no sexual partner was double than with one partner (OR = 2.01, 95% CI = 0.62 to 6.54, p = 0.245). Moreover, participants who had no COVID-19 impact on life were 0.71 times less likely to have high vaccine perception than those who had impact on life (OR = 0.71, 95% CI = 0.372 to 1.36, p = 0.304).

**Table 1. Distribution of sociodemographic profiles among the respondents.**

| Total, N = 408 | n | % |
|---|---|---|
| **Age range (years)** | | |
| 18–25 | 209 | 51.2 |
| 26–35 | 155 | 38.0 |
| 36–45 | 44 | 10.8 |
| **BMI (kg/m$^2$)** | | |
| Below 18.5 | 35 | 8.6 |
| 18.5–25 | 266 | 65.2 |
| Above 25 | 107 | 26.2 |
| **Marital status** | | |
| Unmarried | 252 | 61.8 |
| Married | 156 | 38.2 |
| **Education level** | | |
| Illiterate | 27 | 6.6 |
| Primary | 37 | 9.1 |
| Secondary | 58 | 14.2 |
| Higher Secondary | 117 | 28.7 |
| Graduate/above | 169 | 41.4 |
| **Occupation** | | |
| Service | 129 | 31.6 |
| Business | 55 | 13.5 |
| Self-employed | 41 | 10.1 |
| Student | 145 | 35.5 |
| Unemployed | 23 | 5.6 |
| Others | 15 | 3.7 |
| **Economic status** | | |
| Low | 379 | 92.9 |
| Medium | 20 | 4.9 |
| High | 9 | 2.2 |
| **Residence** | | |
| Urban | 275 | 67.4 |
| Rural | 133 | 32.6 |
| **Living status** | | |
| With family | 311 | 76.2 |
| Without family | 97 | 23.8 |
| **Smoking habit** | | |
| Smoker | 131 | 32.1 |
| Non-smoker | 277 | 67.9 |
| **Sexual orientation** | | |
| Heterosexual | 405 | 99.3 |
| Bisexual | 1 | 0.2 |
| Homosexual | 2 | 0.5 |
| **Sexual partner** | | |
| None | 246 | 60.3 |
| One | 144 | 35.3 |
| More than one | 18 | 4.4 |
| **Covid-19 impact on life** | | |
| Yes | 110 | 27.0 |

(*Continued*)

**Table 1.** (Continued)

| Total, N = 408 | n | % |
|---|---|---|
| No | 298 | 73.0 |

BMI, body mass index; N, number.

## Discussion

As far as we are aware, this is the first study to evaluate the Mpox vaccine perception and vaccination intention of the Bangladeshi adult male population. Significantly, our study findings indicate that education level, age range, marital status, sexual orientation, economic and living status, smoking habit, BMI, residence, Covid-19 impact on life, and the number of sexual partners play a key role in determining the level of vaccine perception and vaccination intention. Most of the respondents showed a higher level of vaccine perception but a medium level of vaccination intention. Moreover, all the sociodemographic factors with Mpox vaccine perception and vaccination intention are found to be significantly associated.

Currently, the incidence of Mpox is dramatically increasing. Originally, Mpox was an endemic zoonotic disease that was restricted to Central and West African countries. Earlier in 2022, in countries where Mpox had never existed before, it began to rise [32]. So, to prevent the spread of the Mpox virus, perception and intention towards vaccination must be strong. According to a similar study in Saudi Arabia, the majority of the participants suggested implementing a preventive measure and to start a vaccination campaign to minimize the spread of Mpox disease. They assessed the vaccination intention with various sociodemographic profiles such as (i) age (18 years and above), (ii) both genders, (iii) marital status (divorced, married, single, widowed), (iv) socioeconomic status, (v) educational level, and (vi) region [32]. A

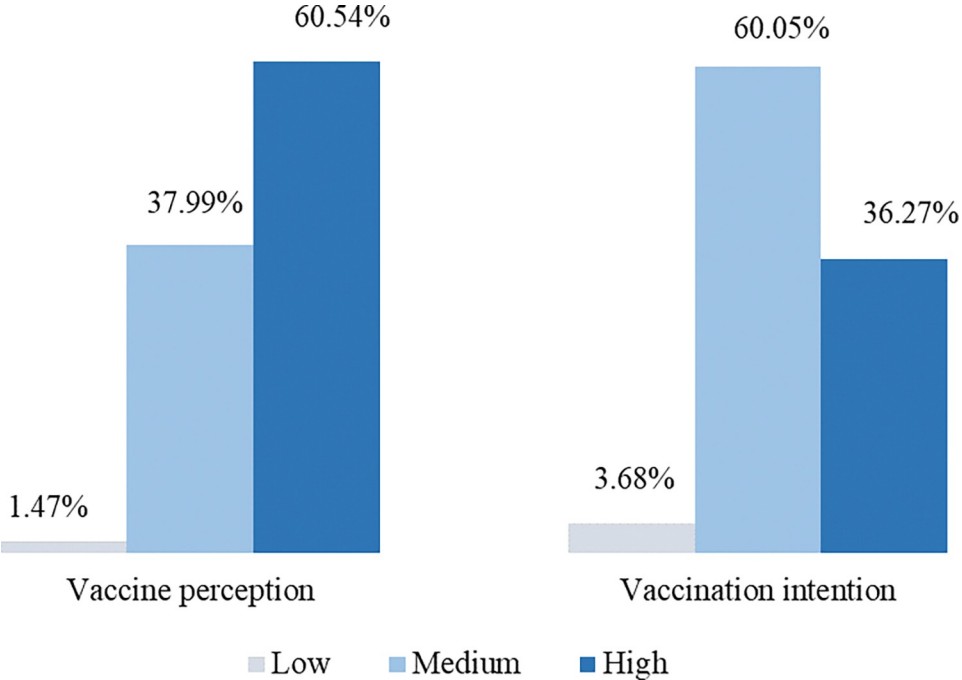

**Fig 1. Level of Mpox (monkeypox) vaccine perception and vaccination intention among the respondents.**

**Table 2. Association between the level of Mpox (monkeypox) vaccine perception and sociodemographic profiles of the respondents.**

| Parameters | Mpox (monkeypox) vaccine perception | | | | | | | | |
| --- | --- | --- | --- | --- | --- | --- | --- | --- | --- |
| | High, N = 247 | | Medium, N = 155 | | Low, N = 6 | | Chi-square tests | | |
| | n | % | n | % | n | % | $\chi^2$ | df | p-value |
| **Age range (years)** | | | | | | | | | |
| 18–25 | 135 | 54.7 | 70 | 45.2 | 4 | 66.7 | 585.378 | 9 | <0.001 |
| 26–35 | 95 | 38.4 | 58 | 37.4 | 2 | 33.3 | | | |
| 36–45 | 17 | 6.9 | 27 | 17.4 | 0 | 0.0 | | | |
| **BMI (kg/m$^2$)** | | | | | | | | | |
| Below 18.5 | 26 | 10.5 | 8 | 5.2 | 1 | 16.6 | 581.914 | 9 | <0.001 |
| 18.5–25 | 160 | 64.8 | 105 | 67.7 | 1 | 16.7 | | | |
| Above 25 | 61 | 24.7 | 42 | 27.1 | 4 | 66.7 | | | |
| **Marital status** | | | | | | | | | |
| Unmarried | 172 | 69.6 | 75 | 48.4 | 5 | 83.3 | 599.316 | 9 | <0.001 |
| Married | 75 | 30.4 | 80 | 51.6 | 1 | 16.7 | | | |
| **Education level** | | | | | | | | | |
| Illiterate | 11 | 4.5 | 16 | 10.3 | 0 | 0.0 | 618.183 | 15 | <0.001 |
| Primary | 17 | 6.9 | 20 | 12.9 | 0 | 0.0 | | | |
| Secondary | 25 | 10.1 | 33 | 21.3 | 0 | 0.0 | | | |
| Higher Secondary | 68 | 27.5 | 45 | 29.0 | 4 | 66.7 | | | |
| Graduate/above | 126 | 51.0 | 41 | 26.5 | 2 | 33.3 | | | |
| **Occupation** | | | | | | | | | |
| Service | 70 | 28.3 | 58 | 37.4 | 1 | 16.7 | 606.049 | 18 | <0.001 |
| Business | 34 | 13.8 | 21 | 13.5 | 0 | 0.0 | | | |
| Self-employed | 19 | 7.7 | 22 | 14.2 | 0 | 0.0 | | | |
| Student | 101 | 40.9 | 39 | 25.2 | 5 | 83.3 | | | |
| Unemployed | 18 | 7.3 | 5 | 3.2 | 0 | 0.0 | | | |
| Others | 5 | 2.0 | 10 | 6.5 | 0 | 0.0 | | | |
| **Economic status** | | | | | | | | | |
| Low | 232 | 93.9 | 141 | 91.0 | 6 | 100.0 | 575.690 | 9 | <0.001 |
| Medium | 8 | 3.2 | 12 | 7.7 | 0 | 0.0 | | | |
| High | 7 | 2.9 | 2 | 1.3 | 0 | 0.0 | | | |
| **Residence** | | | | | | | | | |
| Urban | 186 | 75.3 | 84 | 54.2 | 5 | 83.3 | 596.843 | 9 | <0.001 |
| Rural | 61 | 24.7 | 71 | 45.8 | 1 | 16.7 | | | |
| **Living status** | | | | | | | | | |
| With family | 174 | 70.4 | 131 | 84.5 | 6 | 100.0 | 588.290 | 9 | <0.001 |
| Without family | 73 | 29.6 | 24 | 15.5 | 0 | 0.0 | | | |
| **Smoking habit** | | | | | | | | | |
| Smoker | 69 | 27.9 | 61 | 39.4 | 1 | 16.7 | 583.644 | 9 | <0.001 |
| Non-smoker | 178 | 72.1 | 94 | 60.6 | 5 | 83.3 | | | |
| **Sexual orientation** | | | | | | | | | |
| Heterosexual | 245 | 99.2 | 155 | 100.0 | 5 | 83.3 | 614.832 | 9 | <0.001 |
| Bisexual | 1 | 0.4 | 0 | 0.0 | 0 | 0.0 | | | |
| Homosexual | 1 | 0.4 | 0 | 0.0 | 1 | 16.7 | | | |
| **Sexual partner** | | | | | | | | | |
| None | 170 | 68.9 | 73 | 47.1 | 3 | 50.0 | 597.201 | 9 | <0.001 |
| One | 69 | 27.9 | 73 | 47.1 | 2 | 33.3 | | | |
| More than one | 8 | 3.2 | 9 | 5.8 | 1 | 16.7 | | | |

(*Continued*)

**Table 2.** (Continued)

| Parameters | Mpox (monkeypox) vaccine perception | | | | | | | | |
|---|---|---|---|---|---|---|---|---|---|
| | High, N = 247 | | Medium, N = 155 | | Low, N = 6 | | Chi-square tests | | |
| | n | % | n | % | n | % | χ2 | df | p-value |
| **Covid-19 impact on life** | | | | | | | | | |
| Yes | 81 | 32.8 | 27 | 17.4 | 2 | 33.3 | 584.089 | 6 | **<0.001** |
| No | 166 | 67.2 | 128 | 82.6 | 4 | 66.7 | | | |

p-values are significant at 95% confidence interval (p<0.05). Significant p-values are shown in bold. BMI, body mass index; df, degrees of freedom; N, number.

cross-sectional survey in Pakistan reported that the association between attitude and knowledge with the region of respondents, sex, and type of academic degree toward Mpox were significant [33]. A study in Italy found that the participants had significant knowledge gaps on various factors of Mpox [34]. Another study among 1,352 Italian adults reported that only 26.7% of the respondents were aware about Mpox burden, 47.1% did not have idea about viral transmission, 48.9% were unaware about symptoms, 54.2% were hesitant toward Mpox vaccination, and 38.5% were not confident about vaccines [35]. Nevertheless, respondents of a European study reported higher vaccination intention as they were well aware of the Mpox infection [36]. According to a survey of the general population in the United States, 46% of the respondents were willing to be vaccinated against Mpox [37]. Another study revealed that 77.3% of Indonesian internal medicine residents intended to receive the Mpox vaccine [33]. Initially, many previous studies had shown that there was low vaccination intention among the Bangladeshi population to fight against the Covid-19 virus. More than 50% of the participants were unaware of the effectiveness of the Covid-19 vaccine [38]. Some significant barriers to vaccine hesitancy have been identified, including a lack of faith on the authorities, vaccine-origin conspiracy theories, and misconceptions about religion [39, 40]. Vaccine hesitancy was identified as a serious public health threat by the WHO, raising concerns about the global success of vaccination [39, 41]. Initially, despite having negative perceptions toward a vaccine, later vaccination adequacy increased by creating awareness through several campaigns.

In an Italian study, they had involved 566 physicians only but we involved 408 participants from several occupations [34]. The studies were conducted with both male and female participants in Pakistan and Saudi Arabia but we had performed with only male participants as they are the most affected by the Mpox virus according to several studies [13, 33]. The Pakistani research study involved both healthcare and non-healthcare students as participants. In the European study, large groups of people from different regions of Europe provided the data via smartphone-based online gay-dating apps but we gathered the data from those participants who lived in Bangladesh [42]. However, older respondents of our study as well as European study were more likely to accept vaccinations. According to a similar previous study, the Covid-19 vaccination study was performed among public university students [38]. In both of our studies, the level of education appeared to have a significant impact on the perception and intention toward the vaccine. However, we conducted this study with half of the participants than they did. Moreover, none of the studies assessed the vaccine perception and vaccination intention with sexual orientation, number of sexual partners, and Covid-19 impact on life.

Overall, Bangladeshi participants in this study declared higher perceptions and intentions towards vaccination than other countries such as USA, Europe, and Italy. According to several previous studies regarding the Covid-19 vaccine, the perception and intention of getting injected with the Mpox vaccine rose [13, 33, 34, 38, 42]. Thus, we hope that the Mpox vaccine perception and vaccination intention will also further rise. In Bangladesh, vaccination

**Table 3. Regression analysis of variables by the level of Mpox (monkeypox) vaccine perception among the respondents.**

| | High vaccine perception, N = 247 | | | Medium vaccine perception, N = 155 | | | Low vaccine perception, N = 6 | | |
|---|---|---|---|---|---|---|---|---|---|
| | OR | 95% CI | p-value | OR | 95% CI | p-value | OR | 95% CI | p-value |
| **Age range (years)** | | | | | | | | | |
| 18–25 | 0.694 | 0.223–2.155 | 0.527 | 0.791 | 0.117–5.363 | 0.811 | 0.419 | 0.021–8.540 | 0.572 |
| 26–35 | 1.305 | 0.503–3.384 | 0.585 | 1.161 | 0.227–5.945 | 0.858 | 0.455 | 0.033–6.324 | 0.557 |
| 36–45 | 1 | | | 1 | | | 1 | | |
| **BMI (kg/m²)** | | | | | | | | | |
| 18.5–25 | 0.393 | 0.134–1.151 | 0.088 | 0.537 | 0.096–2.998 | 0.479 | 0.023 | 0.004–0.152 | <0.001 |
| Above 25 | 0.329 | 0.101–1.074 | 0.066 | 0.502 | 0.075–3.356 | 0.477 | 0.349 | 0.053–2.284 | 0.272 |
| Below 18.5 | 1 | | | 1 | | | 1 | | |
| **Marital status** | | | | | | | | | |
| Married | 0.988 | 0.305–3.207 | 0.984 | 0.909 | 0.123–6.697 | 0.925 | 0.002 | 0.000–0.038 | <0.001 |
| Unmarried | 1 | | | 1 | | | 1 | | |
| **Education level** | | | | | | | | | |
| Illiterate | 1.542 | 0.358–6.642 | 0.561 | 1.319 | 0.113–15.381 | 0.825 | 2.701 | 0.005–1411.784 | 0.756 |
| Primary | 1.075 | 0.379–3.048 | 0.892 | 1.044 | 0.174–6.261 | 0.962 | 0.972 | 0.033–28.447 | 0.987 |
| Higher Secondary | 1.999 | 0.877–4.560 | 0.100 | 1.570 | 0.385–6.399 | 0.529 | 8.687 | 0.792–95.263 | 0.077 |
| Graduate/above | 3.718 | 1.518–9.105 | 0.004 | 2.218 | 0.485–10.137 | 0.304 | 1.129 | 0.079–16.124 | 0.929 |
| Secondary | 1 | | | 1 | | | 1 | | |
| **Occupation** | | | | | | | | | |
| Service | 0.434 | 0.122–1.540 | 0.196 | 0.601 | 0.077–4.661 | 0.626 | 0.777 | 0.031–19.582 | 0.878 |
| Business | 0.959 | 0.230–4.001 | 0.954 | 0.950 | 0.093–9.687 | 0.966 | 0.179 | 0.003–9.190 | 0.392 |
| Self-employed | 0.323 | 0.072–1.448 | 0.140 | 0.489 | 0.041–5.769 | 0.570 | 0.114 | 0.001–8.996 | 0.330 |
| Student | 0.641 | 0.180–2.283 | 0.493 | 0.792 | 0.103–6.085 | 0.823 | 11.532 | 0.424–313.367 | 0.147 |
| Others | 0.195 | 0.029–1.289 | 0.090 | 0.335 | 0.015–7.543 | 0.491 | 0.362 | 0.811–1.025 | 0.176 |
| Unemployed | 1 | | | 1 | | | 1 | | |
| **Economic status** | | | | | | | | | |
| Low | 3.336 | 1.031–10.792 | 0.044 | 2.097 | 0.289–15.219 | 0.464 | 2.151 | 0.084–55.140 | 0.643 |
| High | 3.611 | 0.422–30.870 | 0.241 | 2.192 | 0.067–72.159 | 0.660 | 1.658 | 0.007–401.176 | 0.857 |
| Medium | 1 | | | 1 | | | 1 | | |
| **Residence** | | | | | | | | | |
| Urban | 1.919 | 0.163–22.587 | 0.604 | 1.433 | 0.030–68.257 | 0.855 | 0.096 | 0.000–85.245 | 0.498 |
| Rural | 1 | | | 1 | | | 1 | | |
| **Living status** | | | | | | | | | |
| With family | 1.275 | 0.521–3.232 | 0.671 | 1.568 | 0.371–7.390 | 1.395 | 29.641 | 2.944–303.099 | 0.328 |
| Without family | 1 | | | 1 | | | 1 | | |
| **Smoking habit** | | | | | | | | | |
| Non-smoker | 1.030 | 0.557–1.904 | 0.925 | 1.034 | 0.367–2.915 | 0.949 | 2.909 | 0.604–13.997 | 0.183 |
| Smoker | 1 | | | 1 | | | 1 | | |
| **Sexual partner** | | | | | | | | | |
| None | 2.012 | 0.619–6.541 | 0.245 | 1.382 | 0.188–10.183 | 0.751 | 0.432 | 0.051–0.973 | <0.001 |
| More than one | 1.578 | 0.352–7.065 | 0.551 | 1.389 | 0.109–17.657 | 0.800 | 35.868 | 0.469–274.081 | 0.106 |
| One | 1 | | | 1 | | | 1 | | |
| **Covid-19 impact on life** | | | | | | | | | |
| No | 0.712 | 0.372–1.361 | 0.304 | 0.814 | 0.280–2.369 | 0.706 | 0.997 | 0.280–3.545 | 0.996 |
| Yes | 1 | | | 1 | | | 1 | | |

p-values are significant at 95% confidence interval (p<0.05). Significant p-values are shown in bold. BMI, Body Mass Index; CI, Confidence Interval; N, Number; OR, Odds Ratio.

**Table 4. Association between the level of Mpox (monkeypox) vaccination intention and sociodemographic profiles of the respondents.**

| Parameters | Mpox (monkeypox) vaccination intention | | | | | | | | |
|---|---|---|---|---|---|---|---|---|---|
| | High, N = 148 | | Medium, N = 245 | | Low, N = 15 | | Chi-square tests | | |
| | n | % | n | % | n | % | $\chi2$ | df | p-value |
| **Age range (years)** | | | | | | | | | |
| 18–25 | 76 | 51.3 | 123 | 50.2 | 10 | 66.7 | 576.653 | 9 | <0.001 |
| 26–35 | 62 | 41.9 | 90 | 36.7 | 3 | 20.0 | | | |
| 36–45 | 10 | 6.8 | 32 | 13.1 | 2 | 13.3 | | | |
| **BMI (kg/m$^2$)** | | | | | | | | | |
| Below 18.5 | 17 | 11.5 | 16 | 6.5 | 2 | 13.3 | 574.303 | 9 | <0.001 |
| 18.5–25 | 98 | 66.2 | 159 | 64.9 | 9 | 60.0 | | | |
| Above 25 | 33 | 22.3 | 70 | 28.6 | 4 | 26.7 | | | |
| **Marital status** | | | | | | | | | |
| Unmarried | 101 | 68.2 | 141 | 57.6 | 10 | 66.7 | 576.411 | 9 | <0.001 |
| Married | 47 | 31.8 | 104 | 42.4 | 5 | 33.3 | | | |
| **Education level** | | | | | | | | | |
| Illiterate | 3 | 2.0 | 22 | 9.0 | 2 | 13.3 | 614.163 | 15 | <0.001 |
| Primary | 12 | 8.1 | 25 | 10.2 | 0 | 0.0 | | | |
| Secondary | 8 | 5.4 | 46 | 18.8 | 4 | 26.7 | | | |
| Higher Secondary | 46 | 31.1 | 65 | 26.5 | 6 | 40.0 | | | |
| Graduate/above | 79 | 53.4 | 87 | 35.5 | 3 | 20.0 | | | |
| **Occupation** | | | | | | | | | |
| Service | 51 | 34.5 | 73 | 29.8 | 5 | 33.3 | 593.913 | 18 | <0.001 |
| Business | 16 | 10.8 | 39 | 15.9 | 0 | 0.0 | | | |
| Self-employed | 12 | 8.1 | 28 | 11.4 | 1 | 6.7 | | | |
| Student | 55 | 37.1 | 84 | 34.3 | 6 | 40.0 | | | |
| Unemployed | 10 | 6.8 | 13 | 5.3 | 0 | 0.0 | | | |
| Others | 4 | 2.7 | 8 | 3.3 | 3 | 20.0 | | | |
| **Economic status** | | | | | | | | | |
| Low | 138 | 93.2 | 226 | 92.2 | 15 | 100.0 | 578.174 | 9 | <0.001 |
| Medium | 4 | 2.7 | 16 | 6.6 | 0 | 0.0 | | | |
| High | 6 | 4.1 | 3 | 1.2 | 0 | 0.0 | | | |
| **Residence** | | | | | | | | | |
| Urban | 108 | 73.0 | 155 | 63.3 | 12 | 80.0 | 578.038 | 9 | <0.001 |
| Rural | 40 | 27.0 | 90 | 36.7 | 3 | 20.0 | | | |
| **Living status** | | | | | | | | | |
| With family | 102 | 68.9 | 195 | 79.6 | 14 | 93.3 | 580.909 | 9 | <0.001 |
| Without family | 46 | 31.1 | 50 | 20.4 | 1 | 6.7 | | | |
| **Smoking habit** | | | | | | | | | |
| Smoker | 38 | 25.7 | 88 | 35.9 | 5 | 33.3 | 574.733 | 9 | <0.001 |
| Non-smoker | 110 | 74.3 | 157 | 64.1 | 10 | 66.7 | | | |
| **Sexual orientation** | | | | | | | | | |
| Heterosexual | 145 | 98.0 | 245 | 100.0 | 15 | 100.0 | 575.391 | 9 | <0.001 |
| Bisexual | 1 | 0.7 | 0 | 0.0 | 0 | 0.0 | | | |
| Homosexual | 2 | 1.3 | 0 | 0.0 | 0 | 0.0 | | | |
| **Sexual partner** | | | | | | | | | |
| None | 99 | 66.9 | 139 | 56.8 | 8 | 53.3 | 578.441 | 9 | <0.001 |
| One | 47 | 31.8 | 91 | 37.1 | 6 | 40.0 | | | |
| More than one | 2 | 1.3 | 15 | 6.1 | 1 | 6.7 | | | |

(*Continued*)

**Table 4.** (Continued)

| Parameters | Mpox (monkeypox) vaccination intention | | | | | | | | |
|---|---|---|---|---|---|---|---|---|---|
| | High, N = 148 | | Medium, N = 245 | | Low, N = 15 | | Chi-square tests | | |
| | n | % | n | % | n | % | χ2 | df | p-value |
| **Covid-19 impact on life** | | | | | | | | | |
| Yes | 49 | 33.1 | 57 | 23.3 | 4 | 26.7 | 574.320 | 6 | <0.001 |
| No | 99 | 66.9 | 188 | 76.7 | 11 | 73.3 | | | |

p-values are significant at 95% confidence interval (p<0.05). Significant p-values are shown in bold. BMI, body mass index; df, degrees of freedom; N, number.

campaigns have been setup in many locations to create awareness of the disease and prevent the spread of the virus [43–50]. Hence, they have high knowledge about the importance of immunization. So, to get a more effective result, participants from other countries could also follow the same pattern to prevent the spread of the virus among the vast population. Therefore, effective vaccination strategies are necessary. Notably, perception and intention toward different vaccines, along with the degree of misinformation and trust, are likely to change over time; as a result, they need to be evaluated frequently, given their changing nature.

Several limitations apply to the findings in this report. Due to the increased familiarity with the internet and social media, and the proactive attitude toward providing personal details through these platforms, some sub-groups may unintentionally be over-sampled. Similarly, it would be difficult for people to respond if they lack an internet connection in their area, and therefore, some may provide partial information or misinformation. It may be difficult for some of the respondents to understand the questions online, but they may have understood the questions with clarity if interviewed in-person. Also, we included only adult males that might be considered as selection bias of this study. Nonetheless, our study design does have some advantages. Online surveying made it possible to reach a broad spectrum of people in a short time and obtained data that closely matches the characteristics of the larger population that helped to obtain a more conclusive result. By keeping the identity of the participants anonymous or confidential, many participants consequently felt more comfortable providing truthful responses. It became easy for the respondents to complete the online survey form at any time and from any location, whilst the scenario was opposite through in-person data collection. Moreover, we prepared a structured survey questionnaire in English and translated it into Bangla for better comprehension and clarity for the respondents.

To determine the Mpox vaccine perception and vaccination intention among the Bangladeshi male population, a case-control study could be performed in the future to minimize the limitation. More Mpox vaccination research is required to understand the perception and intention of the healthcare professionals such as physicians, pharmacists, and psychologists as they are at a higher risk of exposure.

## Conclusion

Our research study highlighted an incline in vaccine perception and vaccination intention among adult males of Bangladesh to prevent Mpox disease. In groups where there are signs of lower perception and intention, it is important to take steps to increase both the perception and intention of the Mpox vaccination. Additionally, there was a strong correlation between vaccine perception and vaccination intention of the respondents with various sociodemographic profiles such as the level of education, sexual orientation, number of sexual partners, age range, residents living in an urban or rural area, Covid-19 impact on life, smoking habit,

**Table 5. Regression analysis of variables by the level of Mpox (monkeypox) vaccination intention among the respondents.**

| | High vaccination intention, N = 148 | | | Medium vaccination intention, N = 245 | | | Low vaccination intention, N = 15 | | |
|---|---|---|---|---|---|---|---|---|---|
| | OR | 95% CI | p-value | OR | 95% CI | p-value | OR | 95% CI | p-value |
| **Age range (years)** | | | | | | | | | |
| 18–25 | 0.617 | 0.190–2.011 | 0.423 | 0.812 | 0.032–20.357 | 0.899 | 0.910 | 0.106–7.822 | 0.932 |
| 26–35 | 0.883 | 0.038–2.534 | 0.817 | 0.941 | 0.056–15.711 | 0.966 | 0.711 | 0.133–0.686 | 0.689 |
| 36–45 | 1 | | | 1 | | | 1 | | |
| **BMI (kg/m$^2$)** | | | | | | | | | |
| 18.5–25 | 0.695 | 0.301–1.604 | 0.394 | 1.074 | 0.075–15.402 | 0.958 | 1.225 | 0.158–9.488 | 0.846 |
| Above 25 | 0.407 | 0.157–1.054 | 0.064 | 0.906 | 0.048–17.115 | 0.948 | 1.316 | 0.144–12.035 | 0.808 |
| Below 18.5 | 1 | | | 1 | | | 1 | | |
| **Marital status** | | | | | | | | | |
| Married | 0.781 | 0.283–2.150 | 0.632 | 0.894 | 0.051–15.574 | 0.939 | 0.573 | 0.075–4.354 | 0.590 |
| Unmarried | 1 | | | 1 | | | 1 | | |
| **Education level** | | | | | | | | | |
| Illiterate | 0.788 | 0.136–4.558 | 0.790 | 0.952 | 0.013–67.902 | 0.982 | 0.772 | 0.027–22.460 | 0.880 |
| Primary | 2.771 | 0.828–9.276 | 0.098 | 1.690 | 0.073–39.090 | 0.743 | 14.746 | 2.163–100.541 | **0.006** |
| Graduate/above | 5.216 | 1.943–14.001 | **0.001** | 2.015 | 0.159–25.556 | 0.589 | 0.213 | 0.028–1.615 | 0.135 |
| Higher Secondary | 4.249 | 1.676–10.768 | **0.002** | 1.764 | 0.166–18.706 | 0.673 | 0.944 | 0.172–5.170 | 0.947 |
| Secondary | 1 | | | 1 | | | 1 | | |
| **Occupation** | | | | | | | | | |
| Service | 1.162 | 0.416–3.244 | 0.775 | 1.046 | 0.051–21.414 | 0.977 | 6.980 | 0.058–841.881 | 0.427 |
| Business | 0.932 | 0.282–3.084 | 0.909 | 1.009 | 0.032–31.388 | 0.996 | 0.568 | 0.003–93.877 | 0.828 |
| Self-employed | 1.013 | 0.276–3.715 | 0.984 | 1.049 | 0.024–45.356 | 0.980 | 2.966 | 0.017–513.861 | 0.679 |
| Student | 0.726 | 0.267–1.980 | 0.532 | 0.911 | 0.049–17.100 | 0.950 | 5.854 | 0.046–738.038 | 0.474 |
| Others | 0.379 | 0.082–1.756 | 0.215 | 0.484 | 0.128–1.828 | 0.284 | 1.064 | 0.322–3.522 | 0.919 |
| Unemployed | 1 | | | 1 | | | 1 | | |
| **Economic status** | | | | | | | | | |
| Low | 2.668 | 0.771–9.236 | 0.121 | 1.464 | 0.062–34.475 | 0.813 | 5.609 | 0.141–222.888 | 0.359 |
| High | 5.071 | 0.288–89.163 | 0.267 | 1.325 | 0.961–5.815 | 0.956 | 6.681 | 2.877–50.275 | **0.003** |
| Medium | 1 | | | 1 | | | 1 | | |
| **Residence** | | | | | | | | | |
| Urban | 2.785 | 0.387–20.028 | 0.309 | 1.280 | 0.002–720.406 | 0.939 | 0.152 | 0.000–91.323 | 0.564 |
| Rural | 1 | | | 1 | | | 1 | | |
| **Living status** | | | | | | | | | |
| With family | 1.198 | 0.577–2.566 | 0.356 | 1.671 | 0.209–16.408 | 1.739 | 2.488 | 0.514–13.796 | 0.720 |
| Without family | 1 | | | 1 | | | 1 | | |
| **Smoking habit** | | | | | | | | | |
| Non-smoker | 1.122 | 0.608–2.071 | 0.712 | 1.017 | 0.189–5.465 | 0.984 | 1.679 | 0.459–6.134 | 0.433 |
| Smoker | 1 | | | 1 | | | 1 | | |
| **Sexual partner** | | | | | | | | | |
| None | 0.865 | 0.326–2.296 | 0.771 | 0.955 | 0.061–14.966 | 0.974 | 0.502 | 0.067–3.762 | 0.502 |
| More than one | 0.420 | 0.070–2.506 | 0.341 | 0.667 | 0.012–35.869 | 0.842 | 1.587 | 0.205–12.309 | 0.659 |
| One | 1 | | | 1 | | | 1 | | |
| **Covid-19 impact on life** | | | | | | | | | |
| No | 0.867 | 0.496–1.516 | 0.618 | 0.930 | 0.179–4.830 | 0.932 | 0.702 | 0.221–2.232 | 0.548 |
| Yes | 1 | | | 1 | | | 1 | | |

p-values are significant at 95% confidence interval (p<0.05). Significant p-values are shown in bold. BMI, Body Mass Index; CI, Confidence Interval; N, Number; OR, Odds Ratio.

occupation, economic status, BMI, and living status. Nevertheless, our research demonstrated that sociodemographic profiles are significantly related to vaccination hesitancy. As a result, it is essential to create awareness regarding the Mpox vaccine through campaigns. Researchers, practitioners, and policymakers may find our research study helpful for raising the knowledge of the vaccine and vaccination status against Mpox. Adult males of the Bangladeshi population must be immunized as soon as possible to combat this disease.

## Supporting information

**S1 File.**
(XLSX)

## Acknowledgments

All the authors are thankful to the participants for their cooperation in this study.

## Author Contributions

**Conceptualization:** Md. Rabiul Islam, Md. Anamul Haque, Bulbul Ahamed, Md. Tanbir.

**Data curation:** Md. Anamul Haque, Bulbul Ahamed, Md. Tanbir, Md. Robin Khan.

**Formal analysis:** Md. Rabiul Islam, Md. Ashrafur Rahman.

**Investigation:** Md. Robin Khan, Saba Eqbal, Md. Ashrafur Rahman.

**Methodology:** Md. Rabiul Islam, Mohiuddin Ahmed Bhuiyan.

**Project administration:** Mohammad Shahriar, Mohiuddin Ahmed Bhuiyan.

**Supervision:** Md. Rabiul Islam, Mohammad Shahriar, Mohiuddin Ahmed Bhuiyan.

**Writing – original draft:** Md. Rabiul Islam, Saba Eqbal.

**Writing – review & editing:** Md. Rabiul Islam, Mohammad Shahriar, Mohiuddin Ahmed Bhuiyan.

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
