## [Decision Letter · Decision Letter 0]

10 Apr 2023

PONE-D-23-05498Assessment of vaccine perception and vaccination intention of Mpox infection among the adult males in Bangladesh: A cross-sectional study findingsPLOS ONE

Dear Dr. Islam,

Thank you for submitting your manuscript to PLOS ONE. After careful consideration, we feel that it has merit but does not fully meet PLOS ONE’s publication criteria as it currently stands. Therefore, we invite you to submit a revised version of the manuscript that addresses the points raised during the review process.

We look forward to receiving your revised manuscript.

Kind regards,

Academic Editor

PLOS ONE

Journal Requirements:

Reviewers' comments:

Reviewer's Responses to Questions

**Comments to the Author**

1. Is the manuscript technically sound, and do the data support the conclusions?

Reviewer #1: Yes

Reviewer #2: No

2. Has the statistical analysis been performed appropriately and rigorously? 

Reviewer #1: Yes

Reviewer #2: N/A

3. Have the authors made all data underlying the findings in their manuscript fully available?

Reviewer #1: Yes

Reviewer #2: No

4. Is the manuscript presented in an intelligible fashion and written in standard English?

Reviewer #1: Yes

Reviewer #2: No

5. Review Comments to the Author

Reviewer #1: Thank you for the opportunity to review the manuscript "Assessment of vaccine perception and vaccination intention of Mpox infection among the adult males in Bangladesh: A cross-sectional study findings."

It is well-written, and the topic is relevant. I have only minor changes to further improve the quality of this paper.

1. The Material and methods section should better explain what the authors mean with the question regarding the impact of Covid-19 on the respondents’ life.

2. I believe that you should add in the limit of the study that you excluded females of all ages, as this is a clear selection bias.

3. I suggest to compare these results with more experiences that consider not only a possible MPXV vaccination, but the relationship between the general population and monkeypox virus as well, especially in European continent (see doi: https://doi.org/10.3390/pathogens11111285).

Reviewer #2: The authors have conducted a study which is irrelevant in the perspective of country where it is conducted. As of now, no cases of Monkey pox were reported in Bangladesh. Hence, why should one be concerned about vaccination when quick identification, isolation and management would be appropriate to prevent the spread of the disease. And if there is no plan to vaccinate people in near future, why one should be concerned about awareness regarding Mpox vaccine perception and acceptance?

Moreover, the description of result doesn't match with tables and figures.

6. PLOS authors have the option to publish the peer review history of their article (what does this mean?). If published, this will include your full peer review and any attached files.

Reviewer #1: No

Reviewer #2: No

---

## [Author Response · Author response to Decision Letter 0]

10 May 2023

Dear Editors and Reviewers,

Thank you for your letter and the reviewers' comments on our manuscript entitled "Assessment of vaccine perception and vaccination intention of Mpox infection among the adult males in Bangladesh: A cross-sectional study findings" (Manuscript ID PONE-D-23-05498). All the comments were valuable and helpful to the revision and improvement of the manuscript. We have carefully studied the comments and made corrections, which we hope will merit your approval. We marked the revised portions using track changes. Our point-by-point answers to the reviewers’ comments appear at the end of this letter.

We earnestly appreciate the Editors'/Reviewers' work. We hope that after this revision, the paper will be deemed fit for publication. We would be glad to respond to any further questions and comments that you may have. 

Once again, thank you very much for your comments and suggestions.

Best regards,

Md. Rabiul Islam, PhD

Associate Professor, Department of Pharmacy, University of Asia Pacific, 74/A Green Road, Farmgate, Dhaka-1205, Bangladesh. Email: robi.ayaan@gmail.com; Cell: +8801916031831

Point by point authors’ responses to the reviewers

Manuscript ID PONE-D-23-05498

Title: Assessment of vaccine perception and vaccination intention of Mpox infection among the adult males in Bangladesh: A cross-sectional study findings

Reviewer #1

Thank you for the opportunity to review the manuscript "Assessment of vaccine perception and vaccination intention of Mpox infection among the adult males in Bangladesh: A cross-sectional study findings." 

It is well-written, and the topic is relevant. I have only minor changes to further improve the quality of this paper.

Author’s response

Thank you for your review and encouraging comments on our manuscript. We have addressed all your observation very carefully in our revised manuscript. 

Comment 1. The Material and methods section should better explain what the authors mean with the question regarding the impact of Covid-19 on the respondents’ life.

Author’s response

Thank you for your observation. We have now included more information about the impact of Covid-19 on the respondents’ life in the method section for better understanding of readers (page 6, line 19-20). 

Comment 2. I believe that you should add in the limit of the study that you excluded females of all ages, as this is a clear selection bias.

Author’s response

Thank you for your suggestion. We have explained about the inclusion of only males in this study based on the disease epidemiology and prevalence (page 6, lines 3-6).

“Only adult males were invited to participate in this study of ages ranging from 18 to 45 years as this age is vulnerable to the Mpox virus. Moreover, several data studies around the world showed 98% of confirmed cases of Mpox disease among 18-45 years of males. Therefore, we excluded females of all ages.”

However, we have added this point as a limitation of this study to avoid selection biasness (page 18, lines 9-10). 

Comment 3. I suggest to compare these results with more experiences that consider not only a possible MPXV vaccination, but the relationship between the general population and monkeypox virus as well, especially in European continent (see doi: https://doi.org/10.3390/pathogens11111285).

Author’s response

Thank you again for your valuable suggestion. We have now compared our results with the suggested study finding in Europe (page 16, line 21-24). 

Reviewer #2

The authors have conducted a study which is irrelevant in the perspective of country where it is conducted. As of now, no cases of Monkey pox were reported in Bangladesh. Hence, why should one be concerned about vaccination when quick identification, isolation and management would be appropriate to prevent the spread of the disease. And if there is no plan to vaccinate people in near future, why one should be concerned about awareness regarding Mpox vaccine perception and acceptance?

Moreover, the description of result doesn't match with tables and figures.

Author’s response

Thank you for your review and opinion about this manuscript. We have conducted this study regarding Mpox vaccine perception and vaccination intention in Bangladesh where there is no reported case of Mpox infection. Bangladesh is a densely populated lower middle-income country (total population is 169.35 million). The population density in Bangladesh is 1265 per Km2 (3,277 people per mi2). The median age in Bangladesh is 27.6 years. It is difficult to control infectious disease outbreak in such country with large population and weak healthcare infrastructure. Therefore, preventive measures might be a good option for Bangladesh. We conducted this study to report vaccine perception and vaccination intention of Bangladeshi population so that the authorities can know and take necessary measures. Moreover, we totally agree that quick identification, isolation and management are appropriate to prevent the spread of infectious disease. But according to our own experience during Covid-19 pandemic, it was almost impossible to implement these measures in Bangladesh for high population load and literacy level of general people. What Bangladesh did, the country achieved high success in rolling out Covid-19 vaccination (more than 90% people vaccinated). Therefore, the country did face any big threat due to the recent past pandemic. I this context the present study is relevant and useful for the healthcare authorities, policymakers, and others.

Also, we have corrected some mismatch about the description of result with tables and figures in the revised manuscript.

---

## [Editor Report · Decision Letter 1]

15 May 2023

Assessment of vaccine perception and vaccination intention of Mpox infection among the adult males in Bangladesh: A cross-sectional study findings

PONE-D-23-05498R1

Dear Authors,

We’re pleased to inform you that your manuscript has been judged scientifically suitable for publication and will be formally accepted for publication once it meets all outstanding technical requirements.

Kind regards,

The Academic Editor

PLOS ONE

---

## [Editor Report · Acceptance letter]

31 May 2023

PONE-D-23-05498R1 

Assessment of vaccine perception and vaccination intention of Mpox infection among the adult males in Bangladesh: A cross-sectional study findings 

Dear Dr. Islam:

I'm pleased to inform you that your manuscript has been deemed suitable for publication in PLOS ONE. Congratulations! Your manuscript is now with our production department. 

Kind regards, 

on behalf of

Dr. Christian Napoli 

Academic Editor

PLOS ONE